# HTLV-2 Enhances CD8^+^ T Cell-Mediated HIV-1 Inhibition and Reduces HIV-1 Integrated Proviral Load in People Living with HIV-1

**DOI:** 10.3390/v14112472

**Published:** 2022-11-09

**Authors:** María Abad-Fernández, Francisco J. Hernández-Walias, María J. Ruiz de León, María J. Vivancos, María J. Pérez-Elías, Ana Moreno, José L. Casado, Carmen Quereda, Fernando Dronda, Santiago Moreno, Alejandro Vallejo

**Affiliations:** 1Department of Microbiology & Immunology, UNC Chapel Hill School of Medicine, Chapel Hill, NC 27599, USA; 2Laboratory of Inmunovirología, Ramón y Cajal Institute for Health Investigation (IRyCIS), University Hospital Ramón y Cajal, 28034 Madrid, Spain; 3Department of Infectious Diseases, Ramón y Cajal Institute for Health Investigation (IRyCIS), University Hospital Ramón y Cajal, 28034 Madrid, Spain; 4Biomedical Research Center Network in Infectious Diseases (CIBERINFEC), Institute of Health Carlos III (ISCIII), 28029 Madrid, Spain

**Keywords:** HTLV-2, HIV-1, cell-mediated cytotoxicity, HIV-1 reservoir, cell subsets, cytotoxic enzymes

## Abstract

People living with HIV-1 and HTLV-2 concomitantly show slower CD4^+^ T cell depletion and AIDS progression, more frequency of the natural control of HIV-1, and lower mortality rates. A similar beneficial effect of this infection has been reported on HCV coinfection reducing transaminases, increasing the spontaneous clearance of HCV infection and delaying the development of hepatic fibrosis. Given the critical role of CD8^+^ T cells in controlling HIV-1 infection, we analysed the role of CD8^+^ T cell-mediated cytotoxic activity in coinfected individuals living with HIV-1. One hundred and twenty-eight individuals living with HIV-1 in four groups were studied: two groups with HTLV-2 infection, including individuals with HCV infection (N = 41) and with a sustained virological response (SVR) after HCV treatment (N = 25); and two groups without HTLV-2 infection, including individuals with HCV infection (N = 25) and with a sustained virological response after treatment (N = 37). We found that CD8^+^ T cell-mediated HIV-1 inhibition in vitro was higher in individuals with HTLV-2. This inhibition activity was associated with a higher frequency of effector memory CD8^+^ T cells, higher levels of granzyme A and granzyme B cytolytic enzymes, and perforin. Hence, cellular and soluble cytolytic factors may contribute to the lower HIV-1 pre-ART viral load and the HIV-1 proviral load during ART therapy associated with HTLV-2 infection. Herein, we confirmed and expanded previous findings on the role of HTLV-2 in the beneficial effect on the pathogenesis of HIV-1 in coinfected individuals.

## 1. Introduction

Human T cell lymphotropic virus type 2 (HTLV-2) was isolated in 1981 from a patient with atypical hairy cell leukemia [1]. Since then, no clear evidence of human disease has been linked to this retrovirus [2]. HTLV-2 was originally identified in some American Indian populations and Pygmy tribes in Africa [3,4]. It spread rapidly among intravenous drug users (IDUs), mainly in North America and Western Europe [5]. In Spain, the increased use of intravenous drugs among the global population during the 1980s facilitated the transmission of blood-borne pathogens [6,7]. Later, migrants from South America and Eastern Europe changed the distribution and incidence of HTLV-2 in Spain to a lesser extent [8].

It is reported that a person living with HIV-1 has a higher probability of acquiring HTLV-1 or HTLV-2 infection than the general population, as they share mainly the same routes of transmission [9,10,11,12]. In Spain, a non-endemic country, serological screening of blood donation for HTLV is not mandatory, thus the frequency of HIV-1/HTLV is likely underestimated, particularly for HTLV-2, which is rarely associated with disease [13]. Donor suitability is assessed through a pre-donation questionnaire, where individuals from HTLV-endemic countries are deferred from donation. Unfortunately, nationwide surveillance on HTLV prevalence is not available in Spain [14]. Instead, only a few sporadic studies have been reported. A recent study performed in the northeast region of Spain with more than two million blood donors showed only four confirmed HTLV-2 infections (1/500,000), which represents a very low seroprevalence [15]. In a survey performed in our hospital, in Madrid, on 2048 HIV-positive persons, 85 were found to be positive for HTLV-2 (4.15%), of which 81 were both infected with HCV and injecting drug users (95.3%) (data not published). HTLV-2 prevalence among HIV-1-positive persons in our hospital is higher as it serves as a reference hospital for many penitentiary institutions in Madrid. During the 1980s and 1990s, HIV-1 spread rapidly in this closed group of inmates through injecting drugs and was linked to other infections such as HCV and HTLV-2.

The pathogenesis of HTLV-2 is unclear. In a limited number of infected people, HTLV-2 does appear to be associated with a myelopathy/tropical spastic paraparesis (HAM/TSP)-like neurodegenerative disease [16,17,18,19,20]. Unlike HIV-1, which is highly cytopathic, HTLV-2 causes clonal proliferation of the infected cells, mainly CD8^+^ T lymphocytes [21,22]. Yet, HTLV-2 has not been associated with lymphoproliferative disorders [23]. A study investigating HTLV-2 as the etiological agent of large granular lymphocytic leukemia (LGLL) showed that up to 7.5% of LGLL individuals examined were infected with HTLV-2, suggesting its association in a minority of cases. Later, the presence of STAT3 mutations in CD8^+^ T cells harboring HTLV-2, as well as additional mutations in genes involved in JAK/STAT signaling, immune regulation, and lymph proliferation, were described [23]. Nevertheless, no clinical follow-up information was reported to elucidate whether those HTLV-2-positive people carrying those mutations were at increased risk of LGLL.

HTLV-2 impacts both HIV-1 and HCV co-infections. On the one hand, slower CD4^+^ T cell depletion, slower AIDS progression, higher HIV-1 viral control without antiretroviral treatment, and lower mortality rates have been described in people living with HIV-1 [13,24,25,26,27,28,29,30,31]. These effects can be explained, at least in part, by the induction of chemokine and cytokine production, including MIP-1α, MIP-1β, RANTES, and IFN-γ released by HTLV-2-infected CD8^+^ T lymphocytes that can modulate HIV-1 infection and replication in CD4^+^ T lymphocytes [24,32,33,34,35,36,37,38]. These molecules are the natural ligands for CCR5, the main coreceptor for HIV-1 entry into the cells, and can suppress the macrophage-tropic HIV-1 strain infection [35,37]. It has been reported that individuals who remain uninfected despite repeated high-risk sexual exposure to HIV-1 produce high levels of these chemokines in vivo [39]. Moreover, an association between enhanced chemokine production and a slow progression of HIV disease [40], as well as a lower HIV-1 load [41], has been reported. On the other hand, HTLV-2 infection reduces the level of transaminases, increases the spontaneous clearance of HCV infection, and delays the development of hepatic fibrosis in HCV coinfected patients [42,43]. CD8^+^ T cells are critical in the control of viral infection, including HIV-1 and HCV, and are characterized by a high cytotoxic capacity. Effector memory and terminally differentiated CD8^+^ T cell subsets exhibit the highest cytotoxic capacity [44,45], which depends on the lytic granule contents (granzymes and perforin) and the degranulation capacity. To further investigate the potential protective effect of HTLV-2 in individuals living with HIV-1, we compared levels of HIV-1-specific CD8^+^ T cell cytotoxicity activity, evaluated the production of cytolytic enzymes, determined HIV-1 and HTLV-2 proviral load, and determined the distribution of T cell subsets.

## 2. Materials and Methods

### 2.1. Study Design and Participants

This was a cross-sectional study that included 128 Caucasian former intravenous drug users living with HIV-1, performed from 2017 to 2019 in the tertiary Ramon y Cajal University Hospital (Madrid, Spain). These individuals have lived with HIV-1 for a median of 32 years and were under suppressive antiretroviral treatment for 15–19 years at the time of study inclusion. They also were diagnosed for HCV infection at the time they were diagnosed for HIV-1. The serological diagnosis of HTLV-2 of all individuals included in this study was performed during 2015 to 2016 using plasma samples. Four groups were subsequently defined: two groups with HTLV-2 infection, including one group of individuals with HCV infection (with detectable plasma HCV viral load) (N = 41, HT^+^HC^**+**^) and the other with sustained virological response (with undetectable plasma HCV viral load, SVR, after HCV treatment with IFN-γ plus ribavirin) (N = 25, HT^+^HC**^SVR^**); and two control groups without HTLV-2 infection, including one group with HCV infection (with detectable plasma HCV viral load) (N = 25, HT^−^HC^**+**^) and the other group with sustained virological response after treatment with IFN-γ plus ribavirin (N = 37, HT^−^HC**^SVR^**). 

Peripheral blood mononuclear cells (PBMCs) were isolated from EDTA-blood samples by Ficoll–Paque density gradient centrifugation using lymphocyte separation medium (Corning, New York, NY, USA) and cryopreserved in liquid nitrogen. Plasma samples were stored at −80 °C until use.

All individuals included in the study provided written informed consent. This study was conducted following the Declaration of Helsinki (1996) and approved by the institutional review boards of our Hospital Ethics Committee. 

At the moment of study, age, gender, clinical data, history of HIV-1 infection, and antiretroviral therapy were collected from the patient’s records. These data included risk practice for HIV-1 acquisition, time of HIV-1 infection, nadir and current CD4 counts, current HIV-1 viral load and previous therapy, number of previous antiretroviral therapies, and the composition and duration of current cART.

### 2.2. Serological and Virological Assays

HIV-1 antibodies in serum samples were assayed by enzyme immunoassay (EIA, Abbott Laboratories, North Chicago, IL, USA) and reactive results were confirmed by Western blot analysis (DuPont, Wilmington, DE, USA). HIV-1 RNA quantification was measured in plasma samples by quantitative polymerase chain reaction (qPCR, COBAS Ampliprep/COBAS Taqman HIV-1 test, Roche Molecular Systems, Basel, Switzerland) according to the manufacturer’s protocol, with a detection limit of 50 HIV-1 RNA copies/mL.

HCV antibodies were assayed in serum by EIA (Siemens Healthcare Diagnosis, Malvern, PA, USA). HCV RNA quantification was measured in serum by qPCR (COBAS Amplicor, Roche Diagnosis, Barcelona, Spain).

HTLV antibodies were assayed in plasma samples using third-generation Bioelisa HTLV-I + II 5.0 (Biokit, Barcelona, Spain) following the manufacturer’s instructions, and reactive samples were confirmed using Inno-LIA HTLV-I/II Score (Fujirebio Diagnostics, Tokyo, Japan), which discriminates between HTLV-1 and HTLV-2 infection. 

### 2.3. Multiparametric Flow Cytometry Analysis of Lymphocyte T Cell Activation and Subsets

PBMCs were used to analyse CD4^+^ and CD8^+^ T cell subpopulations. Briefly, PBMCs were incubated with the antibodies (Miltenyi Biotec, Bergisch Gladbach, Germany and Beckman Coulter, Brea, CA, USA) for 20 min at 4 °C, washed, and resuspended in PBS containing 1% azide. Cells were analysed in a Gallios Flow Cytometer (Beckman Coulter, Brea, CA, USA). At least 80,000 CD3^+^ T cells were collected for each sample and analysed with Kaluza software (Beckman Coulter), initially gating lymphocytes according to morphological parameters. CD4^+^ and CD8^+^ T cell subpopulations were analysed with the following antibody combination: CD3-allophycocyanin (APC)-A770, CD4-peridinin chlorophyll protein complex (PerCP), CD8-phycoerythrin (PE)-Vio770, CD45RA phycoerythrin (PE) and CCR7-allophycocyanin (APC). The different subpopulations were defined as follows: naïve cells, CD3^+^CD4^+^(CD8^+^)CD45RA^+^CCR7^+^; effector memory T cells (TEM) CD3^+^CD4^+^(CD8^+^)CD45RA^−^CCR7^−^; central memory T cells (TCM) CD3^+^CD4^+^(CD8^+^)CD45RA^−^CCR7^+^; and transitional memory T cells (TEMRA) CD3^+^CD4^+^(CD8^+^)CD45RA^+^CCR7^−^. Activation was defined as the co-expression of CD38-Pacific Blue (PB) and human leucocyte antigen (HLA)-DR-Krome Orange (KrO) on CD4^+^ and CD8^+^ T cells (antibody catalogue number is listed in Appendix A). 

### 2.4. In Vitro CD8^+^ T Cell-Mediated HIV-1 Inhibition

Fresh purified CD8^+^ and CD4^+^ T cells (magnetic cell separation, Miltenyi Biotec, Germany) were used to measure the HIV-1 replication inhibition. HIV-1-NL4.3-Renilla infectious supernatants were obtained from Dr. Alcami, Instituto de Salud Carlos III, Madrid, Spain. HIV-1-NL4.3-Renilla ex vivo infection was performed in purified CD4^+^ T cells alone (control) and autologous CD4/CD8 coculture (ratio 1:1) in 24-well flat-bottom plates, as previously reported [46,47]. CD4^+^ T cells isolated from patients were previously activated with PHA and IL-2 for 72 h. Cells were then infected with 1 ng p24 HIV-1-NL4.3-Renilla per 10^6^ cells for 30 min at gentle rotation and room temperature. After centrifugation at 600× *g* for 30 min at RT, CD8^+^ T cells were added and incubated for 10 days in the presence of IL-2. After measuring the production of renilla in the cell pellets (Renilla Luciferase Assay System, Promega Biotech Iberica, Madrid, Spain) as relative light units (RLUs), cytotoxic activity was calculated according to the following equation: fold cytotoxicity = average renilla (RLUs) in isolated CD4^+^ T cells/average renilla (RLUs) in CD4/CD8 coculture. A value above 1 indicates the grade of viral inhibition accounted for CD8^+^ T cells. Renilla was measured after 5 and 10 days of culture in duplicate. Mean with standard deviation below 0.1 was used to calculate the maximum value of fold cytotoxicity.

### 2.5. Quantification of Plasma Cytolytic Enzymes and Inflammatory Soluble Markers

Granzyme A and granzyme B are enzymes that activate apoptosis when delivered into the target cells in combination with perforin, a pore-forming protein. Their levels were quantified in plasma by Luminex MAGPIX technology (ProcartaPlex Human Basic Kit and ProcartaPlex Human Perforin, Granzyme A, and Granzyme B Simplex). RANTES (CC chemokine ligand 5) has been shown to modulate cytokine production, inducing a switch from Th2-type to Th1-type cytokines. It also inhibits HIV-1 replication because its cellular receptor is CCR5, which is also a coreceptor for HIV-1 (R5 strains). It was quantified by ProcartaPlex Human Basic Kit and ProcartaPlex Human RANTES. IL-6, a pro-inflammatory cytokine, was quantified with ProcartaPlex Human High Sensitivity Basic and ProcartaPlex Human IL-6 High Sensitivity. IL-7, a cytokine important for T cell development and function, was quantified by ProcartaPlex Human Basic Kit and ProcartaPlex Human IL-6.

### 2.6. Integrated HIV-1 Proviral Load Quantification

Genomic DNA from two million total CD4^+^ T cells was isolated using the Gentra Puregene cell kit (Qiagen GmbH, Hilden, Germany) and the DNA was measured using a NanoDrop^TM^ Lite (Thermofisher Scientific, Waltham, MA, USA). An Alu-based quantification of HIV-1 proviral load was performed on genomic DNA using primers encompassing a conserved region in the HIV-1 genome (LTR region) and genomic Alu sequences ProF (5′-AGT AGA TGC TAC GTA ACG TGC TGA ACC CAC TGC TTA AGC CTC) and Alu22R (5′-CTG GGA TTA CAG GCG TGA G) at a concentration of 25 nM each [48,49,50,51]. One microliter aliquots of the outer PCR were subjected to an inner qPCR reaction using primers PVLF (5′-AGT AGA TGC TAC GTA ACG TG) and PVLR (5′-AAG GGT CTG AGG GAT CTC) at a concentration of 25 nM each. Each qPCR reaction was performed in a 10 µL total reaction volume containing Kapa SYBR^®^ FAST mix buffer. The qPCR conditions were 94 °C for 5 min, 40 cycles of 94 °C for 5 s, 57 °C for 15 s, and 63 °C for 45 s. All qPCRs were performed in a 384-well plate format using the LightCycler 480 II (Roche, Basel, Switzerland). In addition, serial dilutions of genomic DNA of an 8E5 cell line (with one HIV-1 copy per cell) generated a standard curve that enabled the quantification of the number of HIV-1 copies per cell in the DNA samples. 

### 2.7. HTLV-2 Proviral Load Quantification

Total HTLV-2 proviral load was quantified by real-time PCR. DNA was extracted from 10^6^ cryopreserved peripheral blood mononuclear cells (PBMCs) using QIAamp-DNA kit (Qiagen-GmbH, Germany). Standard curves were generated in each run using a fivefold dilution of 10^6^ copies of recombinant plasmid (TOPO-TA cloning, Invitrogen) containing one copy of HTLV-2 tax gene fragment (171 bp) and fivefold dilution of 10^6^ copies of a recombinant plasmid containing one copy of human genomic GAPDH gene fragment (162 bp). The amplification reaction was carried out in triplicate using the LightCycler 2.0 (Roche Diagnostics, Basilea, Switzerland). All reactions were performed in 20 µL containing LightCycler FastStart DNA Master PLUS HybProbe 5× (Roche Diagnostics), 200 ng genomic DNA, 50 pmol each primer, TaxF (5′-TACGGTTTTTCCCCAGG) and TaxR (5′-ACTCCTGTCTCCCCCAAG) for tax gene, and GADH-F (5′-CTGACCTGCCGTCTAGA) and GADH-R (5′-GTCGTTGAGGGCAATGC) for GAPDH gene, along with 2 pmol of the following probes: 5′-FAM-CACCCGCCTTCTTCCAATCAATGCGAAAG-TAMRA and 5′-CAGGTGGTCTCCTCTGACTTCAAC-(Flc)/5′-LC705-CGACACCCACTCCTCCACCTTTG-(Phos) for tax gene and GAPDH gene, respectively. PCR conditions for tax and GAPDH genes were as follows: hot start at 95 °C for 10 min, and 40 cycles of denaturation at 95 °C for 10 s, annealing at 62 °C for 15 s, and extension at 72 °C for 20 s. This assay has a detection limit of 50 HTLV-2 copies/10^6^ PBMC.

## 3. Results

### 3.1. Baseline Characteristics of the Study Participants

The baseline characteristics of the patients are shown in Table 1. Gender, time of HIV-1 diagnosis, and time on suppressive ART were similar between the four groups (*p* = 0.193, *p* = 0.396, and *p* = 0.071, respectively (ANOVA test)). Besides, HT^+^ individuals were younger compared with HT^−^ individuals (*p* = 0.001, Student’s *t*-test) and with a shorter time on suppressive ART (*p* = 0.034, Student’s *t*-test). Remarkably, before initiation of cART, plasma HIV-1 RNA was significantly different between the four groups (*p* = 0.039), being more elevated in HT^−^ individuals (*p* = 0.005, Student’s *t*-test) despite having similar nadir CD4^+^ T cell count (*p* = 0.324, ANOVA test). No significant differences were found in the level of plasma HCV RNA between HT^+^HC^+^ and HT^−^HC^+^ groups (*p* = 0.481). CD4^+^ T cell count and percentages were different between the four groups (*p* = 0.002 and *p* = 0.001, respectively), being higher in the HT^+^HC^svr^ and HT^−^HCV^svr^ groups. CD8^+^ T cell count was similar between the four groups (*p* = 0.548), whereas the percentages were different (*p* = 0.021), being lower in the HT^+^HC^svr^ and HT^−^HC^svr^ groups. The differences led to significant differences in the CD4/CD8 ratio between the groups (*p* = 0.002), showing a higher CD4/CD8 ratio in the HT^+^HC^svr^ and HT^−^HC^svr^ groups, but with no differences between the HT^+^ and HT^−^ groups.

### 3.2. CD8^+^ T Cell-Mediated HIV-1 Inhibition Was Higher in HTLV-2 Coinfected Individuals

CD8^+^ T cell-mediated inhibition of the replication of HIV-1-NL4-3-Renilla in CD4/CD8 cocultures was significantly higher in HT^+^ (N = 13) compared with HT^−^ (N = 20) individuals (*p* = 0.005) (Figure 1). Moreover, the inhibition was higher in the HT^+^HC^+^ group (N = 8) compared with the HT^−^HC^svr^ group (N = 17) (*p* = 0.013).

The CD8^+^ T cell-mediated HIV-1 inhibition assay was performed only with the luciferase-based assay because of the high correlation between RLU values and other assay values for the detection of HIV-1 replication such as the quantification of HIV-p24 or retrotranscriptase activity [52,53].

### 3.3. HIV-1 Proviral Load in CD4^+^ T Cells Was Lower in HTLV-2 Coinfected Individuals

CD4^+^ T cells isolated from HTLV-2 coinfected individuals (N = 42) harbored significantly lower HIV-1 DNA copies than HTLV-2 negative individuals (N = 19) (*p* = 0.039) (Figure 2). According to the HCV infection status, both HT^+^HC^+^ and HT^+^HC^svr^ individuals had lower levels of HIV-1 proviral load compared with either the HT^−^HC^+^ or HT^−^HC^svr^ groups, although they were significant only between the HT^+^HC^+^ and HT^−^HC^svr^ groups (*p* = 0.009).

### 3.4. Higher Levels of Granzyme A, Granzyme B, Perforin, RANTES, and IL-6 in HTLV-2 Coinfected Individuals

Higher levels of granzyme A, granzyme B, and perforin were found in HT^+^ individuals (*p* < 0.001, *p* < 0.001, and *p* < 0.001, respectively) compared with HT^−^ individuals. Higher levels were also found in accordance with the HCV infection status (Figure 3). The levels of RANTES and IL-6 were also higher in HT^+^ individuals (*p* < 0.001 and *p* < 0.001, respectively) compared with HT^−^ individuals. Again, the levels were higher in HT^+^ regardless of HCV infection status (Figure 3). Nevertheless, the level of IL-7 was similar comparing HT^+^ and HT^−^ individuals regardless of the HCV infection status.

### 3.5. Higher Effector Memory CD8^+^ T Cells and Lower CD8^+^ T Cell Activation in HTLV-2 Individuals

We then evaluated CD4^+^ and CD8^+^ T cell subsets in all individuals (Figure 4 and Figure 5, respectively). While no differences were found in the CD4^+^ T cell subsets between HT^+^ and HT^−^ individuals (Figure 4), a statistically significant higher level of effector memory CD8^+^ T cells was found in HT^+^ compared with HT^−^ individuals (*p* < 0.001) (Figure 5). These higher levels were balanced with a reduction in the levels of naïve (*p* = 0.024) and central memory (*p* = 0.070) CD8^+^ T cell subsets. Instead, the level of TEMRA (transitory) CD8^+^ T cell subset was similar between HT^+^ and HT^−^ individuals. These higher levels of effector memory CD8^+^ T cells were also found regardless of the HCV infection status. The presence of HCV infection did not alter the level of this cell subset in either HT^+^ or HT^−^ individuals. Moreover, HCV infection did not alter the level of either naïve or central memory CD8^+^ T cells. 

On the other hand, while a similar CD4^+^ T cell activation was found between HT^+^ and HT^−^ individuals, HCV infection lead to higher levels of activation in both HT^+^ and HT^−^ individuals compared with HC^svr^ individuals (*p* = 0.023 and *p* = 0.001, respectively) (Figure 6). However, HTLV-2 associated with lower CD8^+^ T cell activation was found in HT^+^ individuals, although not significant (*p* = 0.067). The presence of HCV infection increased the level of activation compared with HC^svr^ individuals. 

### 3.6. CD8^+^ T Cell-Mediated HIV-1 Inhibition Positively Correlated with Effector Memory CD8^+^ T Cell Subset, Cytolytic Enzymes, and Perforin

CD8^+^ T cell cytotoxic capacity positively correlated with the percentage of effector memory CD8^+^ T cells (*p* = 0.006) and levels of perforin (*p* = 0.010), granzyme A (*p* = 0.017), and granzyme B (*p* < 0.001) (Figure 7). No correlation was found, however, either with HIV-1 proviral load, RANTES, IL-6, IL-7, naïve CD8^+^ T cells, CD8^+^ TCM, or CD8^+^ TEMRA (data not shown).

### 3.7. Integrated HIV-1 Proviral Load Negatively Correlates with RANTES 

HIV-1 proviral load negatively correlated with the level of RANTES (*p* = 0.014) (Figure 8). No correlation was found, however, with cytolytic enzymes, perforin, IL-6, IL-7, or any CD4^+^ or CD8^+^ T cell subsets (data not shown).

### 3.8. HTLV-2 Proviral Load Positively Correlated with the Effector Memory CD8^+^ T Cell Subset and Granzyme A

HTLV-2 proviral load positively correlated with the effector memory CD8^+^ T cell subset (*p* = 0.025) and the level of granzyme A (*p* = 0.045) (Figure 9). No correlation was found, however, with cytolytic enzymes, perforin, IL-6, IL-7, HIV-1 viral load, or any CD4^+^/CD8^+^ T cell subsets (data not shown).

## 4. Discussion

HTLV-2 impacts both HIV-1 and HCV coinfections. A slower CD4^+^ T cell depletion and progression to AIDS and an increase in natural HIV-1 control with lower HIV-1 viral load has been reported in HTLV-2/HIV-1 coinfected patients in the absence of antiretroviral treatment [16,25,26,27,28,54]. On the other hand, it has been reported that HTLV-2 infection reduces the level of transaminases, increases the spontaneous clearance of HCV infection, and delays the development of hepatic fibrosis in HTLV-2/HIV-1/HCV coinfected patients [31,42,43]. Given that CD8^+^ T cells are critical in the control of HIV replication, as evidenced by the increase in HIV-specific CD8^+^ T cells with the decrease in viremia in acutely infected people [55,56], we investigated the impact of HTLV-2 on the ability of CD8^+^ T cells to inhibit HIV-1 replication in autologous CD4^+^ T cells in vitro and mediated cytolytic enzymes’ production.

While it has been reported that HIV-specific T cell responses are maintained at relatively high frequencies in long-term antiretroviral-mediated viral-suppressed individuals, the proliferative and cytotoxic capacity of HIV-specific CD8^+^ T cells is not fully restored [57,58].

In the current study, we found that the ability of CD8^+^ T cells to mediate HIV-1 inhibition in vitro was higher in individuals coinfected with HTLV-2. If HIV-1 and HTLV-2 coinfection occurred simultaneously, HTLV-2 may be preventing the decrease in cytotoxic potential of HIV-1 specific CD8^+^ T cells along with disease progression and/or favoring the restoration of their cytotoxicity during antiretroviral therapy. The former could be supported by the fact that HTLV-2 coinfected individuals showed a lower plasma HIV-1 viral load before the initiation of the antiretroviral treatment, confirming previous observations [16,54]. In parallel with this finding, Bassani et al. reported that HIV-1-Gag-specific CD8^+^ T cell response, identified by the release of IFN-γ and MIP-1β, was higher in antiretroviral-naïve HIV-1HTLV-2 coinfected individuals compared with HIV-1 monoinfected individuals [59]. Moreover, these authors could associate this higher response with a lower HIV viral load. However, in contrast with our study, they did not mention the HCV status of the studied individuals.

Unlike HIV-1, which is highly cytopathic, HTLV-2 causes clonal proliferation of the infected cells, mainly CD8^+^ T lymphocytes [12,13]. Therefore, the higher cytotoxic potential of HIV-1-specific CD8^+^ T cells associated with HTLV-2 may be a result of increased numbers of antigen-specific cells within the augmented total CD8^+^ T cells. In this regard, high frequencies of functionally competent circulating tax-specific CD8^+^ T cells were previously detected in HTLV-2 monoinfected individuals [60], indicating that cytotoxic activity is also directed against HTLV-2. With this trend of thought, we would expect a lower HCV viral load in HTLV-2 coinfected individuals, but instead, we found a similar level when compared with HTLV-2 uninfected individuals. Higher CD8^+^ T cell-mediated HIV-1 inhibition positively correlated with the frequency of effector memory CD8^+^ T cells and cytolytic activity in HTLV-2 coinfected individuals. Higher levels of IL-6 were associated with HTLV-2, which is required to maintain the cytolytic function [61]. Elevated IL-6 levels in parallel with a lower level of activation (C reactive protein) have been reported in another similar HIV scenario [62,63]. The authors found this effect in HIV-1 elite controllers (undetectable viral load without antiretroviral treatment) compared with infected individuals under antiretroviral treatment. They also found an increased HIV-1-Gag-specific CD8^+^ T cell response in HIV-1 elite controllers [63]. The authors did not test the presence of HTLV-2 infection in these individuals; however, we performed a study in which HTLV-2 coinfection was found in nearly 20% of individuals with natural control of HIV-1 replication (unpublished data). Hence, multiple cellular and soluble factors may be contributing to inhibiting the replication of HIV-1. 

Interestingly, we observed a positive correlation between effector memory CD8^+^ T cells and HTLV-2 proviral load. This highly differentiated memory CD8^+^ T cell subset might be the most susceptible to HTLV-2 infection that undergoes a clonal expansion upon recognition of antigen peptide. Further studies in which HTLV-2 proviral load is measured in different T cell subsets would be necessary to elucidate which subset is more susceptible for HTLV-2 infection.

We also found that the lower level of integrated HIV-1 proviral load in HTLV-2-seropositive individuals was directly associated with increased levels of RANTES, confirming previous results [35]. The chemokine RANTES is a natural ligand of CCR5, one of the major HIV-1 co-receptors, potentially inhibiting infection by CCR5-dependent HIV-1 isolates in coinfected individuals [35,37].

One limitation of this study was the limited number of HT^+^ individuals from whom we could obtain fresh blood samples to perform the in vitro assay. Another limitation was that the γδ T cell subset and NK cells, both with known high unspecific cytotoxic activity that could play an important role in the weakening of these viral infections [64], were not analyzed. In addition, we were not able to collect any HTLV-2/HIV-1 coinfected individuals to analyze the effect of the HCV infection, as individuals with suppressive viral response cannot be taken as HCV negative. 

Besides, HIV/HCV coinfection leads to enhanced HIV-1 disease progression [65], and in this study, we observed that, in the HIV-1 positive group with the HCV infection, HTLV-2 was able to control or inhibit the HIV-1 replication successfully, although there is the presence of HCV infection, suggesting that HTLV-2 exerts a strong encountering effect over the HIV-1 replication.

In summary, we confirmed and expanded current knowledge on the beneficial effect of HTLV-2 on the pathogenesis of HIV-1 in coinfected people. We found higher CD8^+^ T cell-mediated HIV-1 inhibition activity in vitro, associated with a higher frequency of effector memory CD8^+^ T cells and higher levels of cytolytic granzymes and perforin ex vivo, along with reduced levels of integrated proviral HIV-1.

## Figures and Tables

**Figure 1 viruses-14-02472-f001:**
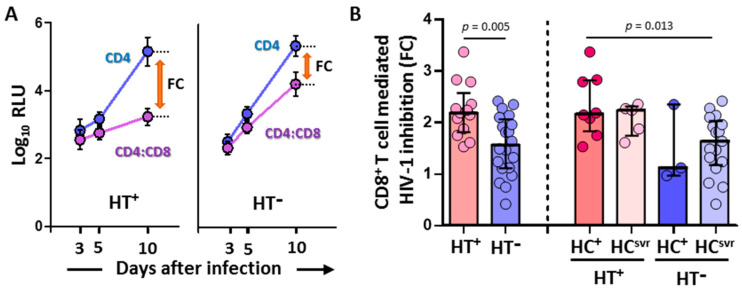
(**A**) Representative samples showing relative light units (log_10_ RLU) at 3, 5, and 10 days after infection in CD4 alone and CD4/CD8 coculture (mean and standard deviation are shown for each value). Fold cytotoxicity (FC) is shown for each of the HT^+^ and HT^−^ samples. (**B**) CD8^+^ T cell-mediated HIV-1 inhibition in HIV-1-infected individuals with the following coinfections: HT^+^, HTLV-2 positive; HT^−^, HTLV-2 negative; HC^+^, HCV positive; and HC^svr^, treated for HCV with sustained virological response. Mann–Whitney U test. Only statistically significant values are shown. Significant when *p* < 0.05.

**Figure 2 viruses-14-02472-f002:**
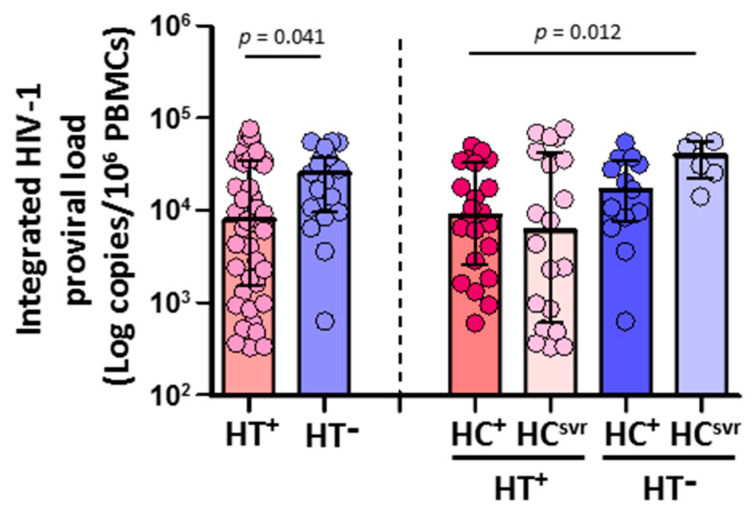
Integrated HIV-1 proviral load in HIV-1-infected individuals with the following coinfections: HT^+^, HTLV-2 positive; HT^−^, HTLV-2 negative; HC^+^, HCV positive; and HC^svr^, treated for HCV with sustained virological response. Mann–Whitney U test. Only statistically significant values are shown. Significant when *p* < 0.05.

**Figure 3 viruses-14-02472-f003:**
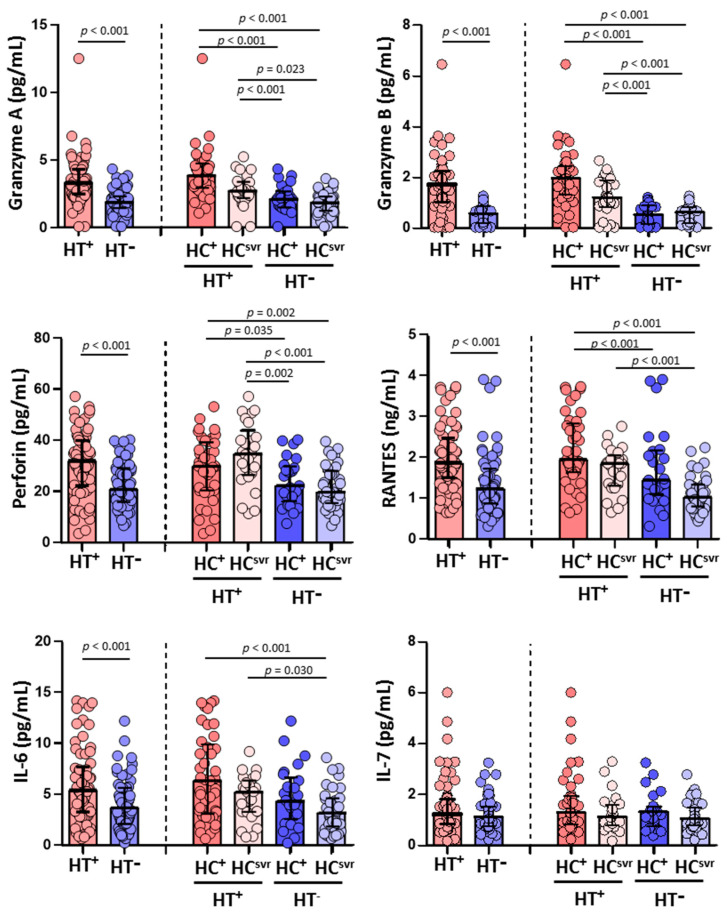
Levels of granzyme A, granzyme B, perforin, RANTES, IL-6, and IL-7 in HIV-1-infected individuals with the following coinfections: HT^+^, HTLV-2 positive; HT^−^, HTLV-2 negative; HC^+^, HCV positive; and HC^svr^, treated for HCV with sustained virological response. Mann–Whitney U test. Only statistically significant values are shown. Significant when *p* < 0.05.

**Figure 4 viruses-14-02472-f004:**
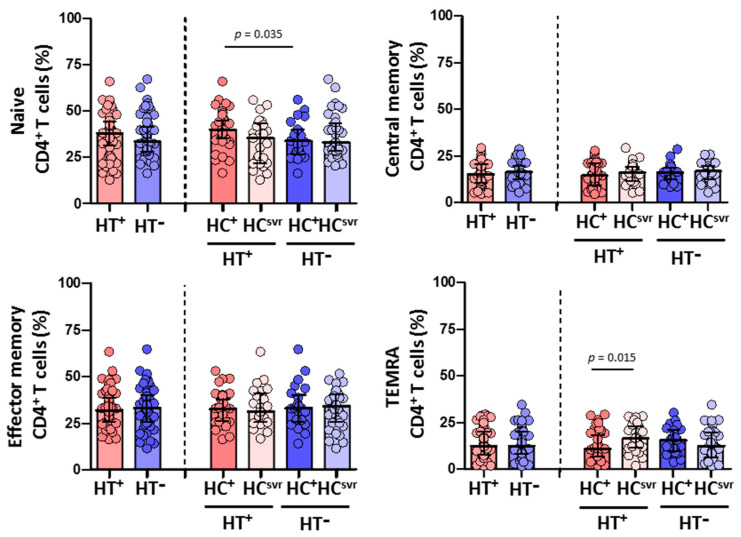
Percentage of CD4^+^ T cell subsets: naïve, central memory, effector memory, and TEMRA in HIV-1-infected individuals with the following coinfections: HT^+^, HTLV-2 positive; HT^−^, HTLV-2 negative; HC^+^, HCV positive; and HC^svr^, treated for HCV with sustained virological response. Mann–Whitney U test. Only statistically significant values are shown. Significant when *p* < 0.05.

**Figure 5 viruses-14-02472-f005:**
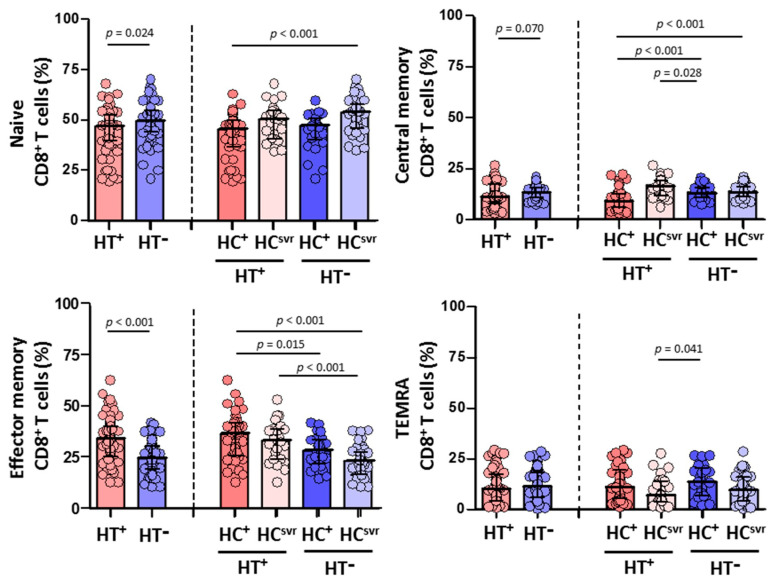
Percentage of CD8^+^ T cell subsets: naïve, central memory, effector memory, and TEMRA in HIV-1-infected individuals with the following coinfections: HT^+^, HTLV-2 positive; HT^−^, HTLV-2 negative; HC^+^, and HCV positive; HC^svr^, treated for HCV with sustained virological response. Mann–Whitney U test. Only statistically significant values are shown. Significant when *p* < 0.05.

**Figure 6 viruses-14-02472-f006:**
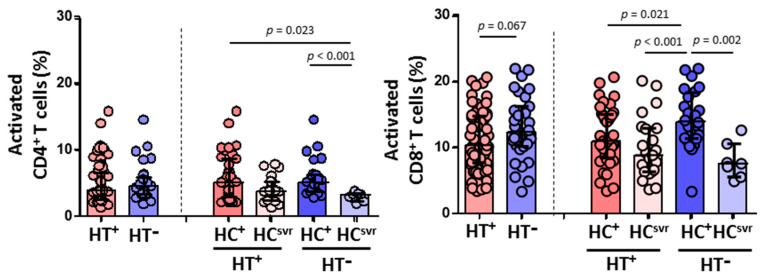
Activation levels of CD4^+^ and CD8^+^ T cells in HIV-1-infected individuals with the following coinfections: HT^+^, HTLV-2 positive; HT^−^, HTLV-2 negative; HC^+^, HCV positive; and HC^svr^, treated for HCV with sustained virological response. Mann–Whitney U test. Only statistically significant values are shown. Significant when *p* < 0.05.

**Figure 7 viruses-14-02472-f007:**
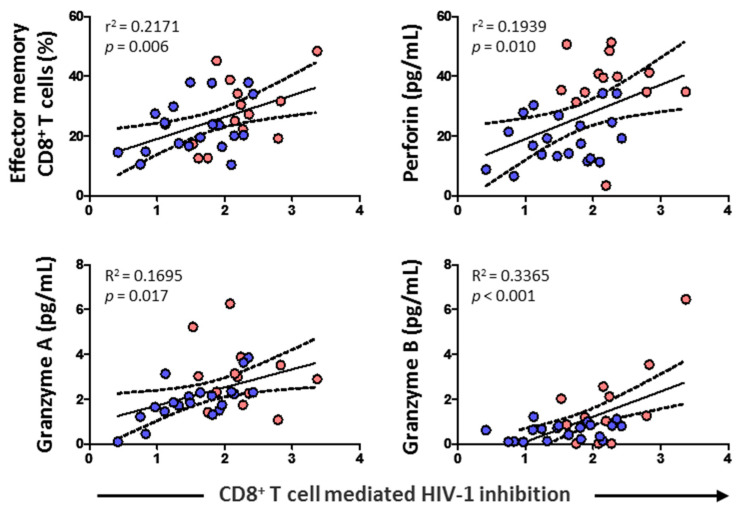
Association of CD8^+^ T cell-mediated HIV-1 inhibition with effector memory CD8^+^ T cell percentage, and the levels of perforin and granzymes in HIV-1-infected individuals with HTLV-2 infection (red dots) and without HTLV-2 infection (blue dots). Spearman correlation. Significant when *p* < 0.05.

**Figure 8 viruses-14-02472-f008:**
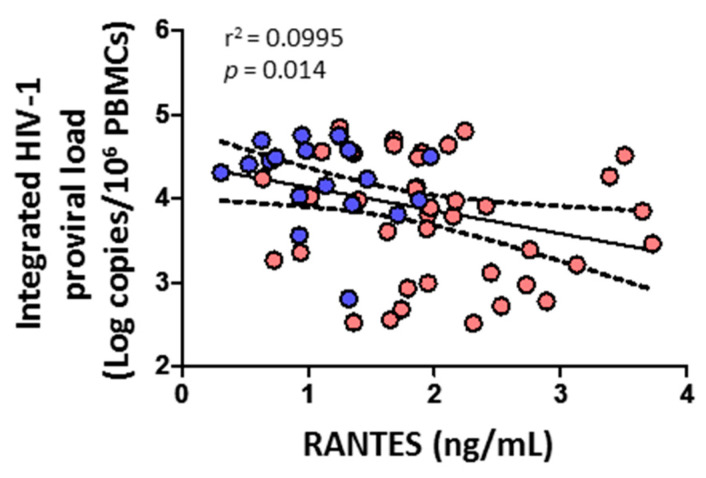
Association of integrated HIV-1 proviral load with soluble chemokine RANTES in HIV-1-infected individuals with HTLV-2 infection (red dots) and without HTLV-2 infection (blue dots). Spearman correlation. Significant when *p* < 0.05.

**Figure 9 viruses-14-02472-f009:**
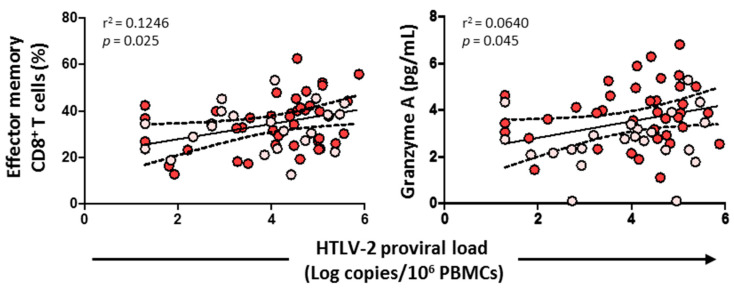
Association of HTLV-2 proviral load with effector memory CD8^+^ T cells and granzyme A levels in HIV-1-infected individuals with HTLV-2 infection as well as with HCV infection (red dots) and HCV treated with sustained virological response (pink dots). Spearman correlation. Significant when *p* < 0.05.

**Table 1 viruses-14-02472-t001:** Characteristics of the HIV-1-infected participants according to their HTLV-2 and HCV co-infection status.

	HT^+^ Group		HT^−^ Group		ANOVA	Student’s *t*-Test
	HC^+^ Group	HC^svr^ Group	HC^+^ Group	HC^svr^ Group	4 Groups	HT^+^ vs. HT^−^
N	41	25	25	37		
Age (years)	44 (43–49)	45 (42–48)	47 (43–51)	51 (47–52)	**0.001**	**0.001**
Gender (male)	29 (70.7%)	19 (76%)	23 (92%)	26 (70.3%)	0.193	0.409
Time of HIV-1 diagnosis (months)	384 (343–415)	371 (319–415)	381 (353–410)	389 (347–427)	0.396	0.359
Time on suppressive ART (months)	186 (167–210)	197 (171–221)	196 (179–231)	234 (222–281)	0.071	**0.034**
Pre-ART plasma HIV-1 RNA (log copies/mL)	4.7 (4.2–5.2)	4.7 (3.95–5)	4.9 (4.7–5.3)	4.8 (4.58–5.35)	**0.039**	**0.005**
Nadir CD4^+^ T cell count (cells/mm^3^)	128 (67–189)	160 (55–276)	114 (44–164)	114 (37–204)	0.324	0.165
CD4^+^ T cell count (cells/mm^3^)	441 (297–574)	502 (343–667)	411 (214–542)	649 (519–814)	**0.002**	0.080
Percentage	21 (14–30)	25 (22–32)	20 (14–25)	28 (24–37)	**0.001**	0.316
CD8^+^ T cell count (cells/mm^3^)	954 (633–1410)	925 (794–1011)	872 (775–1280)	814 (574–1107)	0.584	0.605
Percentage	53 (40–59)	45 (39–57)	50 (40–56)	40 (31–48)	**0.021**	0.062
CD4^+^/CD8^+^ ratio	0.39 (0.26–0.69)	0.61 (0.43–0.81)	0.40 (0.28–0.63)	0.74 (0.49–1.18)	**0.002**	0.309
Plasma HCV RNA (log copies/mL)	6.11 (5.59–6.61)	Und	6.43 (5.78–6.89)	Und	-	0.481

HT^+^HC^+^ group, HTLV-2/HIV-1/HCV coinfected individuals; HT^+^HC^svr^ group, HTLV-2/HIV-1 coinfected individuals with sustained virological response after HCV treatment; HT^−^HC^+^ group, HIV-1/HCV coinfected individuals; HT^−^HC^svr^, HIV-1 coinfected individuals with sustained virological response after HCV treatment; ART, antiretroviral treatment; Und, undetectable. Statistical significance in bold when *p* < 0.05. Median and interquartile range are shown.

## Data Availability

The original data presented in the study can be seen upon request to the corresponding author.

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
