# Peer review of "HTLV-2 Enhances CD8+ T Cell-Mediated HIV-1 Inhibition and Reduces HIV-1 Integrated Proviral Load in People Living with HIV-1"

_viruses, 2022, doi:10.3390/v14112472_

Round 1

Reviewer 1 Report

HTLV-2 is a frequent viral co-pathogen associated with HIV and is reported to exhibit hinder HIV-1 replication. As CD8 cells play critical role in controlling HIV-1 infection, authors studied CD8+ T cell-mediated cytotoxicity in individuals co-infected with HTLV-2, HIV-1 and HCV using combination of in vitro and ex vivo experiments to achieve the aim of the study. The study strongly supported the previously reported finding that HTLV-2 negatively affects HIV-1, with a correlation of the chemokine (RANTES) in controlling HIV-1 during the HIV-1/HTLV-2 co-infection. They also observed that CD8+ T cell-mediated HIV-1 inhibition was higher in HTLV-2 individuals which was found to be associated with increased number/levels of effector memory CD8+ T cells, granzyme A and granzyme B and perforin, indicating the  beneficial effect of HTLV-2 on the pathogenesis of HIV-1 in co-infected individuals. This is an important aspect which suggests the suitability of HTLV-2/HIV-1 co-infection as a model for studying the newer strategic approaches for HIV-1 vaccine development. The paper is well written however there are some suggestions which will enhance the quality of the paper

Introduction:

1.       Looking at the number of co-infected individuals in the study, it looks like that HIV-1, HCV and HTLV-2 infections are common in Spain. Reviewing the available data, a small paragraph explaining the prevalence of individual as well as co-infecting viral pathogens may be added in the Introduction section.

2.       Introduction should be more focused. Needs to be elaborated focusing on the background behind the study carried out, aim of the study, etc.

3.       Reference no- 3 and 4 do not support the sentence. New references may be added or the sentence may be avoided. 

Material and methods

1.       Authors mention that this is a retrospective study. It is not clear if the presented study is a part of any previous study or it is a different study. If it is different, the original study may be explained briefly with an ethical clearance statement. Additionally, a statement mentioning approval of the institutional ethics committee for the use of stored samples from the previous study needs to be added.

2.       Adequate explanation of the groups (control?), HT+/HT-HCV+ and with a sustained virological response (SVR) after HCV treatment (IFNγ + ribavirin) is needed for a better understanding.

3.       The period of patient enrollment and sample collection is not mentioned.

4.       L 85-86: These individuals have had undetectable HIV-1 viral load under antiretroviral treatment for at least one year before the study inclusion. Reframe the sentence.

5.       L 123: supernatants were gently obtained from Dr. Alcami, Instituto de Salud Carlos III

The word gently may be removed. Obtained or gifted looks more appropriate

6.       Check the Forward primer used for the Pol gene (Line no -155)? It might be for LTR. Correct it accordingly.

7.       There is no information on how diagnosis and confirmation for HTLV-2 was carried out.

Results

1.       Table 1 focuses only on the HCV and HIV-1 viral loads, CD4 and CD8 counts but does not mention about HTLV-2 diagnosis or confirmation.

2.       CD8+ T cell-mediated HIV-1 inhibition: Confirmation of results of the luciferase assay is essential.

3.       It would be good if authors further identify expression of the HIV-1 proteins by western blot or other techniques

4.       L212: Moreover, the inhibition was higher in the HT+HC+ group (N=8) compared to the HT-HCsvr (N=17) (p=0.011). Is it HT+HC+ groupSVR group?

5.       L 246-251: While no differences were found in the CD4+ T cell subsets between HT+ and HT- individuals (Figure 4), a statistically significant higher level of effector memory CD8+ 247 T cells was found in HT+ compared to HT- individuals (p<0.001) (Figure 5). These higher levels were balanced with a reduction in the levels of naïve (p=0.023) and central memory (p=0.070) CD8+ T cell subsets. Instead, the level of TEMRA (transitory) CD8+ T cell subset was similar between HT+ and HT- individuals.   

If there is a significant difference in the levels of effector memory cells of HT+ and HT- individuals, how the level of TEMRA can be similar? Explain.

Discussion:

1.       Needs improvement with a discussion of the results obtained in the study and comparing them with the results of the other studies.

2.       L 309-311: It has been reported a slower CD4+ T cell depletion and progression to AIDS, an increase in HIV-1 control (undetectable HIV-1 viral load in the absence of antiretroviral treatment), and lower HIV-1 viral load in HTLV-2/HIV-1 co-infected patients [14,23-26,47]. Reframe the sentence. Something is wrong.  Also how all viral loads can be undetectable?

3.       It looks like that the line 314 discusses the results obtained in this study, focusing on the HCV infection. Reframe the sentence.

4.       L 338-340: With this trend of thought, we would expect a lower HCV viral load in HTLV-2 co-infected individuals, but instead, we found a similar level compare to HTLV-2 uninfected individuals.

Should be “as/when compared to”

5.       L348-349: Further studies in which HTLV-2 proviral load is measured in different T cell subsets would be necessary to elucidate it. Reframe the sentence.

6.       L352-354: The chemokine RANTES is a natural ligand of CCR5, one of the major HIV-1 co-receptors, potentially inhibiting infection by CCR5-dependent HIV-1 isolates in co-infected individuals. Add reference

Author Response

Introduction:

Looking at the number of co-infected individuals in the study, it looks like that HIV-1, HCV and HTLV-2 infections are common in Spain. Reviewing the available data, a small paragraph explaining the prevalence of individual as well as co-infecting viral pathogens may be added in the Introduction section.

Response: The following has been included in the Introduction: “In Spain, a non-endemic country, serological HTLV screening of blood donation is not mandatory. Donor suitability is assessed through a pre-donation questionnaire, where individuals from HTLV-endemic countries are deferred from donation. Unfortunately, nationwide surveillance on HTLV prevalence is not available in Spain. Instead, only a few local studies have been reported. In one recent HTLV seroprevalence study performed in Catalonia, northeast Spain, in more than 2 million blood donors, only four were confirmed as HTLV-2 positive (1/500000). In a survey performed in our hospital, Madrid, on 2.048 HIV-positive persons, 85 resulted in HTLV-2 positive (4.8%). Of them, 81 were infected with HCV and injecting drug users (95.3%). An explanation for this elevated prevalence among HIV-1-positive persons in our hospital is the fact that it serves as a reference hospital for many penitentiary institutions in Madrid. During the ´80s and ´90s, HIV-1 spread rapidly in this closed group of inmates through injecting drugs. And the HIV-1 spread was linked to other infections such as HCV and HTLV-2”.

Introduction should be more focused. Needs to be elaborated focusing on the background behind the study carried out, aim of the study, etc.

Response: Introduction has been modified.

Reference no- 3 and 4 do not support the sentence. New references may be added or the sentence may be avoided. 

Response: I apologize for the mistake. Reference #3 has been replaced by Ref #4 (Vandamme et al, 1998). Ref #4 is correct though, it refers to the origin and distribution of HTLV-2 in the Americas (now is Ref #4). 

Material and methods

Authors mention that this is a retrospective study. It is not clear if the presented study is a part of any previous study or it is a different study. If it is different, the original study may be explained briefly with an ethical clearance statement. Additionally, a statement mentioning approval of the institutional ethics committee for the use of stored samples from the previous study needs to be added.

Response: This was a prospective cross-sectional study performed from 2017 to 2019. The mistake was fixed.

Adequate explanation of the groups (control?), HT+/HT-HCV+ and with a sustained virological response (SVR) after HCV treatment (IFNγ + ribavirin) is needed for a better understanding.

Response: The groups have been better explained.

The period of patient enrollment and sample collection is not mentioned.

Response: Fixed

L 85-86: These individuals have had undetectable HIV-1 viral load under antiretroviral treatment for at least one year before the study inclusion. Reframe the sentence.

Response: Reframed

L 123: supernatants were gently obtained from Dr. Alcami, Instituto de Salud Carlos III. The word gently may be removed. Obtained or gifted looks more appropriate.

Response: Fixed

Check the Forward primer used for the Pol gene (Line no -155)? It might be for LTR. Correct it accordingly.

Response: Corrected

There is no information on how diagnosis and confirmation for HTLV-2 was carried out.

Response: Added

Results

Table 1 focuses only on the HCV and HIV-1 viral loads, CD4 and CD8 counts but does not mention about HTLV-2 diagnosis or confirmation.

Response: We added that information in Materials and Methods (participants).

CD8+ T cell-mediated HIV-1 inhibition: Confirmation of results of the luciferase assay is essential. It would be good if authors further identify expression of the HIV-1 proteins by western blot or other techniques.

Response: Further analysis is not possible since we do not have stored supernatant aliquots. However, in other projects, we compared luciferase versus HIV-p24 assays and both gave a positive correlation. Hence, we decided to focus on luciferase assay for this study since we are so confident about the results. On the other hand, we performed the assay in duplicate and at two different time points, 5 and 10 days after infection. If during the assay, the only-CD4 wells gave no production of renilla, the experiment was not included in the study even if the production of renilla were detected in the CD4:CD8 wells.  

L212: Moreover, the inhibition was higher in the HT+HC+ group (N=8) compared to the HT-HCsvr (N=17) (p=0.011). Is it HT+HC+ groupSVR group?

Response: We compare HT+HC+ groups to the HT-HCsvr group.

L 246-251: While no differences were found in the CD4+ T cell subsets between HT+ and HT- individuals (Figure 4), a statistically significant higher level of effector memory CD8+ T cells was found in HT+ compared to HT- individuals (p<0.001) (Figure 5). These higher levels were balanced with a reduction in the levels of naïve (p=0.023) and central memory (p=0.070) CD8+ T cell subsets. Instead, the level of TEMRA (transitory) CD8+ T cell subset was similar between HT+ and HT- individuals. If there is a significant difference in the levels of effector memory cells of HT+ and HT- individuals, how the level of TEMRA can be similar? Explain.

Response: The increase of effector memory T cells is compensated with a decrease of naïve and central memory T cells. It is not necessary to alter the level of TEMRA cells to have these 4 subsets balanced.

Discussion:

Needs improvement with a discussion of the results obtained in the study and comparing them with the results of the other studies.

Response: Discussion has been modified.

L 309-311: It has been reported a slower CD4+ T cell depletion and progression to AIDS, an increase in HIV-1 control (undetectable HIV-1 viral load in the absence of antiretroviral treatment), and lower HIV-1 viral load in HTLV-2/HIV-1 co-infected patients [14,23-26,47]. Reframe the sentence. Something is wrong.  Also how all viral loads can be undetectable?

Response: The sentence has been reframed. HIV-1 viral loads are undetectable in all persons included in the study because they all are under suppressive antiretroviral treatment.  The other way to have an undetectable HIV-1 viral load is when the person can naturally control the HIV-1 replication in the absence of antiretroviral treatment. These subjects are called HIV controllers.

It looks like that the line 314 discusses the results obtained in this study, focusing on the HCV infection. Reframe the sentence.

Response: They are variables that were reported elsewhere regarding HCV coinfection. The sentence has been reframed.

L 338-340: With this trend of thought, we would expect a lower HCV viral load in HTLV-2 co-infected individuals, but instead, we found a similar level compare to HTLV-2 uninfected individuals. Should be “as/when compared to”

Response: Fixed

L348-349: Further studies in which HTLV-2 proviral load is measured in different T cell subsets would be necessary to elucidate it. Reframe the sentence.

Response: Reframed

L352-354: The chemokine RANTES is a natural ligand of CCR5, one of the major HIV-1 co-receptors, potentially inhibiting infection by CCR5-dependent HIV-1 isolates in co-infected individuals. Add reference

Response: Added

Reviewer 2 Report

In this manuscript, the authors evaluated the beneficial effect of HTLV-2 on HIV-1 infection. After the analyses of 128 HIV-1 positive patients coinfected with HTLV-2 as well as HCV, some key parameters concerning the HIV-1-related immunity, such as CD4/CD8 subsets and CD8 activation, have been statistically calculated. Data showed that HTLV-2 coinfection may facilitate CD8 T cell-mediated HIV-1 inhibition, which was also evidenced by low HIV-1 proviral load in CD4 T cells. Overall, this study is conceptually interesting by providing new insight into the potential upregulation of CD8 T cell-mediated cytotoxic activity in HTLV-2/HIV-1 coinfected patients. However, all samples collected for study were HCV positive (SVR cannot be simply viewed as HCV negative, even virus is undetectable after treatments), and the existence of HCV may interfere with the conclusion. If HCV negative samples (only HTLV-2/HIV-1 coinfection) were collected, the effect of HTLV-2 on HIV-1 infection might be different. Above issue needs to be taken into consideration, at least the authors should discuss it in manuscript.

Following points also need to be clarified.

1 Some p values are inconsistent between manuscript and figures, for example in Fig 1, the p values are 0.004 and 0.011 in manuscript, while in the figure, they are 0.005 and 0.013. Please verify all figures and fix this issue.

2 Some comparisons and statistics between groups are meaningless, for example in lines 212-213, HT+HC+ vs HT-HCsvr. Same issues are also seen in Fig 2, 3, 5 and 6.

Author Response

In this manuscript, the authors evaluated the beneficial effect of HTLV-2 on HIV-1 infection. After the analyses of 128 HIV-1 positive patients coinfected with HTLV-2 as well as HCV, some key parameters concerning the HIV-1-related immunity, such as CD4/CD8 subsets and CD8 activation, have been statistically calculated. Data showed that HTLV-2 coinfection may facilitate CD8 T cell-mediated HIV-1 inhibition, which was also evidenced by low HIV-1 proviral load in CD4 T cells. Overall, this study is conceptually interesting by providing new insight into the potential upregulation of CD8 T cell-mediated cytotoxic activity in HTLV-2/HIV-1 coinfected patients. However, all samples collected for study were HCV positive (SVR cannot be simply viewed as HCV negative, even virus is undetectable after treatments), and the existence of HCV may interfere with the conclusion. If HCV negative samples (only HTLV-2/HIV-1 coinfection) were collected, the effect of HTLV-2 on HIV-1 infection might be different. Above issue needs to be taken into consideration, at least the authors should discuss it in manuscript.

Response: This is a good point. Actually, we do not have any HTLV-2/HIV-1 coinfected patients to perform the comparison. This comment has been added in Discussion.

Following points also need to be clarified.

Some p values are inconsistent between manuscript and figures, for example in Fig 1, the p values are 0.004 and 0.011 in manuscript, while in the figure, they are 0.005 and 0.013. Please verify all figures and fix this issue.

Response: Fixed

Some comparisons and statistics between groups are meaningless, for example in lines 212-213, HT+HC+ vs HT-HCsvr. Same issues are also seen in Fig 2, 3, 5 and 6.

Response: We wanted to show all significant differences, including those comparing people with detectable HCV infection with people with undetectable HCV viral load after sustained virological response.

Reviewer 3 Report

The article by Abad-Fernadez et.al have analyzed the role of CD 8+ mediated cytotoxic activity in people living with co infection (HTLV-2 or HCV) and found that HTLV-2 infection can be beneficial for people with HIV . A lot of work has being done, however in the article few of experimental details are missing. All the participants used in this study were HIV positive, but no data shown here to confirm the results. Proviral load or ELISA test is not provided for patients. Only integrated proviral load in CD4+ cells given. Patient samples are only used for all experiments not healthy donors are compared in results or patients only with HIV-1.

In the materials and methods- Under Flow cytometry analysis and In-vitro CD 8+ T cell activation, the methods are not clearly written and please add catalog number of antibodies used

Table 2- please check alignments

Author Response

The article by Abad-Fernandez et.al have analyzed the role of CD 8+ mediated cytotoxic activity in people living with co infection (HTLV-2 or HCV) and found that HTLV-2 infection can be beneficial for people with HIV . A lot of work has being done, however in the article few of experimental details are missing. All the participants used in this study were HIV positive, but no data shown here to confirm the results. Proviral load or ELISA test is not provided for patients. Only integrated proviral load in CD4+ cells given. Patient samples are only used for all experiments not healthy donors are compared in results or patients only with HIV-1.

Response: How patients were diagnosed with HIV-1 has been added in Materials and Methods. HIV-1 viral load before antiretroviral treatment is shown in table 1 (All were positive). Also, they were all reactive with ELISA. We wanted to compare the groups of patients with the most similarities, hence we matched patients for HIV and HCV infections, and injecting drug use. Thereby, HTLV infection is the only different variable to analyze different effects produced by this virus.

In the materials and methods- Under Flow cytometry analysis and In-vitro CD8+ T cell activation, the methods are not clearly written and please add catalog number of antibodies used

Response: Added as supplementary material

Table 2- please check alignments

Response: Fixed

Round 2

Reviewer 1 Report

There are many grammatical errors. Check carefully. Following are my comments on the revised manuscript

     Abstract:

Line No 33: Should be “we analysed the role of”

Introduction:

·         Line Nos 62-65: Avoid repetition. Both sentences mean the same.

·         Line No 64-65: Correct it to “serological screening of blood donors for HTLV is not mandatory”

·         Line No 68-69: Correct it to “instead, only a few sporadic studies have been reported. In a recent study performed in Catalonia”

·         Line no.71-72: 85 were found to be positive for HTLV-2 (4.8%), of which 81 were injecting drug users (95.3%) and infected with HCV (data not published).

·         Line Nos 73-76: prevalence among HIV-1-positive persons in our hospital is higher as it serves as a reference hospital for many penitentiary institutions in Madrid. During the ´80s and ´90s, HIV-1 spread rapidly in this closed group of inmates through injecting drugs and was linked to other infections such as HCV and HTLV-2.

·         Line No 94: Should be “in CD4+ T lymphocytes”

 Material and Methods:

·      Line Nos 116-118: Authors state that these individuals have lived with HIV-1 for a median of 32 years and were under suppressive antiretroviral treatment for 15-19 years at the time of study inclusion. Please correct. I appreciate the efforts of the national ART programme.

·         Line No 118: Correct it to “They also were diagnosed for HCV infection at the time they were diagnosed for HIV-1.”  

·         Line No 119-120: The serological diagnosis for HTLV-2 of all individuals included in this study was performed during 2015 to 2016 using plasma samples.

·         Line No 140: Should be “HIV antibodies in serum samples were assayed by -----“

·         Line no. 142: For HIV-1 quantitation, plasma samples are recommended. Please recheck “whether Serum or plasma was used”. It should be “HIV-1 RNA quantification was measured in plasma samples”

·         Line no. 151: Should be “confirmed using Inno-LIA HTLV-I/II Score (Fujirebio Diagnostics, Japan), that discriminates”

Results:

Following comment is not addressed well.

 ·         CD8+ T cell-mediated HIV-1 inhibition: Confirmation of results of the luciferase assay is essential. It would be good if authors further identify expression of the HIV-1 proteins by western blot or other techniques.

o   Not fully convinced with the provided reasoning. Considering the unavailability of stored supernates, it would be ideal if authors add their published reference where they have shown positive correlation of the luciferase versus HIV-p24 assays. Or a subset of the data from the previous project can be added to the manuscript if it is not published.

o   There is no information on how many assays were considered for statistical analysis. No error bars are shown in the figures and standard deviation is not provided. 

Discussion:

 ·         Line Nos 382-386: Needs to be reframed like this. A slower CD4+ T cell depletion and progression to AIDS, an increase in natural HIV-1 control with a lower HIV-1 viral load has been reported in HTLV-2/HIV-1 coinfected patients in the absence of antiretroviral treatment [16,25-28,52].

·         Line 441 to 444: Instead of discussing the study this way, authors may discuss it as follows

The HIV-HCV coinfection leads to enhanced HIV-1 disease progression (provide ref.) and in this study, we observed that the in HIV-1 positive group with the HCV infection, HTLV-2 was able to control or inhibit the HIV-1 replication successfully although there is presence of HCV infection, suggesting that HTLV-2 exerts strong encountering effect over the HIV-1 replication.

Author Response

Please, see attached file.

Reviewer 2 Report

The authors have addressed all my concerns and the revised manuscript could be considered for publication.

Author Response

The authors have addressed all my concerns and the revised manuscript could be considered for publication.

Response: We thank the reviewer for her/his comments and suggestions.

Reviewer 3 Report

In the article about How HTLV-2 infection reduces the infection as well as proviral load in people with HIV infection, the authors have made a great effort in putting together the study and shown that how CD 8 cells play a role in controlling HIV infection in people co infected with HTLV-2 and HIV .

Just few minor changes has to be made in the introduction

Please check the % of HTLV 2 positive patients in Line 71.

check line number 68- 70, 73-78, please reframe the sentence

Author Response

In the article about How HTLV-2 infection reduces the infection as well as proviral load in people with HIV infection, the authors have made a great effort in putting together the study and shown that how CD 8 cells play a role in controlling HIV infection in people co infected with HTLV-2 and HIV .

Just few minor changes has to be made in the introduction

Please check the % of HTLV 2 positive patients in Line 71.

Response: Percentage has been changed to 4.15%

check line number 68- 70, 73-78, please reframe the sentence

Response: Sentences have been reframed.

We want to thank the reviewer for her/his comments and suggestions.